# Construction of All-in-Focus Images Assisted by Depth Sensing

**DOI:** 10.3390/s19061409

**Published:** 2019-03-22

**Authors:** Hang Liu, Hengyu Li, Jun Luo, Shaorong Xie, Yu Sun

**Affiliations:** 1School of Mechatronic Engineering and Automation, Shanghai University, No. 99 Shangda Road BaoShan District, Shanghai 200444, China; liuhang@shu.edu.cn (H.L.); luojun@shu.edu.cn (J.L.); srxie@shu.edu.cn (S.X.); sun@mie.utoronto.ca (Y.S.); 2Department of Mechanical and Industrial Engineering, University of Toronto, Toronto, ON M5S 3G8, Canada

**Keywords:** all-in-focus, image fusion, depth sensing

## Abstract

Multi-focus image fusion is a technique for obtaining an all-in-focus image in which all objects are in focus to extend the limited depth of field (DoF) of an imaging system. Different from traditional RGB-based methods, this paper presents a new multi-focus image fusion method assisted by depth sensing. In this work, a depth sensor is used together with a colour camera to capture images of a scene. A graph-based segmentation algorithm is used to segment the depth map from the depth sensor, and the segmented regions are used to guide a focus algorithm to locate in-focus image blocks from among multi-focus source images to construct the reference all-in-focus image. Five test scenes and six evaluation metrics were used to compare the proposed method and representative state-of-the-art algorithms. Experimental results quantitatively demonstrate that this method outperforms existing methods in both speed and quality (in terms of comprehensive fusion metrics). The generated images can potentially be used as reference all-in-focus images.

## 1. Introduction

The depth of field (DoF) of an imaging system is limited. With a fixed focus setting, only objects in a particular depth range appear focused in the captured source image, whereas objects in other depth ranges are defocused and blurred. An all-in-focus image in which all objects are in focus has many applications, such as digital photography [1], medical imaging [2], and microscopic imaging [3,4]. A number of all-in-focus imaging methods have been proposed, which can be grouped into two categories: point spread function (PSF)-based methods and RGB-based multi-focus image fusion methods.

The PSF-based methods obtain an all-in-focus image by estimating the PSF of the imaging system and restoring an all-in-focus image based on the estimated PSF. A partially-focused image can be modelled as an all-in-focus image convolved with a PSF. Deconvolution methods first estimate the PSF and then deconvolve with this PSF to restore an all-in-focus image. The PSF of a partially-focused image is non-uniform because the farther an object is from the DoF of an imaging system, the larger is the extent of blurriness of the object in an image. One type of deconvolution method directly estimates the non-uniform PSF of an imaging system using specially-designed cameras [5] or a camera with a specially-designed lattice-focal lens [6]. Instead of estimating the non-uniform PSF, the other type of deconvolution method first constructs an image with uniform blur and then estimates a uniform PSF. The image with uniform blur can be obtained by scanning the focus positions [4,7] or moving the lens or image detector [8] during a single detector exposure. The wave-front coding technique is another approach to obtain a uniform blur image by adding a suitable phase mask to the aperture plane and making the optical transfer function of the imaging system defocus invariant [9,10,11,12]. The deconvolution methods enable single-shot extended DoF imaging. However, deconvolution ringing artefacts can appear in the resulting image, and high frequencies can be captured with lower fidelity [8].

In RGB-based multi-focus image fusion methods, in-focus image blocks are distinguished from among multiple multi-focus source images that are captured using different focus settings, to construct an all-in-focus image. Existing multi-focus image fusion algorithms include multi-scale transform [13,14], feature space transform [15,16], spatial domain methods [2,17,18,19,20], pulse coupled neural networks [21,22], and deep convolutional neural networks [23].

In multi-focus image fusion, one challenge is to obtain a reference all-in-focus image, which better reflects the ground truth, to which other methods are directly compared. Due to the lack of reference images, a number of metrics were defined for indirectly comparing the performance across multi-focus image fusion methods. As discussed in [24], various metrics, such as information theory-based metrics, image feature-based metrics, image structural similarity-based metrics, and human perception-based metrics [25], were developed because they all represent different aspects of the quality of an all-in-focus image.

In order to obtain a reference all-in-focus image, if the distances between all objects and the camera are known, the in-focus image blocks can be directly determined by choosing those objects whose distances are within the DoF of the camera. This is enabled by the advent and rapid advances of depth sensors (e.g., Microsoft Kinect and ZED stereo camera), which provide a convenient approach for accurately determining the distances of objects in a scene.

Actually, depth maps from depth sensors and colour images from traditional cameras are complementary to each other. Depth maps provide depth information of objects, which have been integrated with colour images to improve the performance of object tracking [26], the resolution of colour images [27], the detection of perspective-invariant features [28], etc. Compared with colour images, the resolution of depth maps from consumer depth sensors is lower with much noise. Thus, colour images have also been used to improve the resolution of depth maps and reduce the noise [29,30,31].

In this paper, our idea is to use the depth information from a depth map to assist the fusion of multiple multi-focus source images to construct an all-in-focus image. To our knowledge, this is the first work to use a depth map to help solve the all-in-focus imaging problem. Instead of distinguishing in-focus image blocks from among multi-focus source images, a graph-based depth map segmentation algorithm is proposed to directly obtain in-focus image block regions by segmenting the depth map. The distances of objects in each segmented in-focus image block region are confined to be within the DoF of the camera such that all objects in the region appear focused in a multi-focus source image. These regions are used to guide the focus algorithm to locate an in-focus image for each region from among multi-focus source images to construct an all-in-focus image. Experimental results quantitatively demonstrate that this method outperforms existing methods in both speed and quality (in terms of fusion metrics); thus, the generated images can potentially be used as reference all-in-focus images. The proposed method is not dependent on a specific depth sensor and can be implemented with structured light-based depth sensors (e.g., Microsoft Kinect v1), time of flight-based depth sensors (e.g., Microsoft Kinect v2), stereo cameras (e.g., ZED stereo camera), and laser scanners.

## 2. Multi-Focus Image Fusion System

In Figure 1, the image detector is at a distance of *v* from a lens with focal length of *f*. A scene point *M*, at a distance of *u* from the lens, is imaged in focus at *m*. If the lens moves forward with a distance of *p* from the lens, then *M* is imaged as a blurred circle centred around $m^{\prime }$, while the scene point *N* at a distance of $u^{\prime }\;\left(u^{\prime }<u\right)$ from the lens is imaged in focus at *n*. In optics, if the distance between *m* and $m^{\prime }$ is less than the radius of the circle of confusion (CoC) in the image plane, all the scene points between *M* and *N* appear acceptably sharp in the image. This indicates that by changing the distance between the lens and image detector while capturing images, objects at different distance ranges appear focused in order in the captured multi-focus source images. DoF can be divided into back DoF (denoted by b_DoF in this work) and front DoF (denoted by f_DoF in this work), which indicate the depth range of objects after and before the precisely in-focus scene point that can appear acceptably sharp in an image.

Figure 2a shows our multi-focus image fusion system, which consists of a focus-tunable Pentax K01 colour camera with an 18–55-mm lens and a Kinect depth sensor. The diameter of the CoC of the colour camera δ is 0.019 mm; the aperture value *F* was set to 4.0; and the focal length *f* was set to 24 mm. The flowchart of the proposed multi-focus image fusion method is shown in Figure 2b. In this method, the depth map and multi-focus source images of an unknown scene are captured using the Kinect depth sensor and Pentax colour camera, respectively. Then, the depth map is segmented into multiple in-focus image block regions, and the objects in each region are within a DoF and all appear focused. These segmented in-focus image block regions are used to guide the focus algorithm to locate an in-focus image from among multi-focus source images for each region. Finally, the all-in-focus image is constructed by combining the in-focus images of all segmented regions.

## 3. Detailed Methods

Figure 3 uses an example to illustrate the main steps and intermediate results when the proposed multi-focus image fusion method is applied to construct an all-in-focus image of a scene. Firstly, the depth map from the Kinect depth sensor is preprocessed to align with the colour image captured with the colour camera, based on a stereo calibration method, and to recover the missing depth values. A graph-based image segmentation algorithm is then used to segment the preprocessed depth map into regions. A focus algorithm is used to locate an in-focus image for each region from among multi-focus source images to construct an all-in-focus image.

### 3.1. Depth Map Preprocessing

#### 3.1.1. Align Depth Map with Colour Image

Microsoft Kinect contains a depth sensor and an RGB camera that provides both depth and colour streams with a resolution of 640 × 480 at 30 Hz. The depth sensor consists of an infrared (IR) projector combined with an IR camera. The IR projector projects a set of IR dots, and the IR camera observes each dot and matches it with a dot in the known projector pattern to obtain a depth map. The operating range of the present Kinect depth sensor is between 0.5 m and 5.0 m [32].

Due to the different spatial positions and intrinsic parameters of the IR camera of the Kinect depth sensor and of the Pentax colour camera, the depth map is not aligned with the colour image. To align the depth map with the colour image, the depth map is first mapped to 3D points in the IR camera’s coordinate system using the intrinsic parameters of the IR camera. Then, these 3D points are transformed to the Pentax colour camera’s coordinate system using extrinsic parameters that relate the IR camera’s coordinate system and the colour camera’s coordinate system. Finally, the transformed 3D points are mapped to the colour image coordinate system using the intrinsic parameters of the colour camera.

Let $\left(u_{0},v_{0}\right)$ denote the coordinates of the principal point of the IR camera, $f_{x}$ and $f_{y}$ denote the scale factors in the image *u* and *v* axes of the IR camera, and $u_{0}$, $v_{0}$, $f_{x}$, and $f_{y}$ be the intrinsic parameters of the IR camera. Let [u,v,Z] represent a pixel in the depth map, *Z* represent the depth value in [u,v], and ${\left[X,Y,Z\right]}^{T}$ represent the mapped 3D point of [u,v] in the IR camera coordinate system. According to the pinhole camera model, the values of *X* and *Y* can be calculated according to:(1)$$\begin{array}{c} X=\left(u-u_{0}\right)Z/f_{x},\\ Y=\left(v-v_{0}\right)Z/f_{y}. \end{array}$$

Let *R* and *T* represent the rotation and translation that relate the coordinate system of the IR camera of the Kinect depth sensor and the colour camera’s coordinate system. *R* and *T* are the extrinsic parameters. *R* is a 3×3 matrix, and *T* is a 3×1 matrix. The relationship between the transformed 3D point ${\left[X^{\prime },Y^{\prime },Z^{\prime }\right]}^{T}$ in the colour camera’s coordinate system and ${\left[X,Y,Z\right]}^{T}$ can be expressed as:(2)$${\left[X^{\prime },Y^{\prime },Z^{\prime }\right]}^{T}=R{\left[X,Y,Z\right]}^{T}+T.$$

Let $\left(u_{0}^{\prime },v_{0}^{\prime }\right)$ denote the coordinates of the principal point of the colour camera and ${f_{x}}^{\prime }$ and ${f_{y}}^{\prime }$ denote the scale factors in the image $u^{\prime }$ and $v^{\prime }$ axes of the colour camera. After mapping ${\left[X^{\prime },Y^{\prime },Z^{\prime }\right]}^{T}$ to the colour image coordinate system, the aligned depth point $\left[u^{\prime },v^{\prime },Z^{\prime }\right]$ can be obtained, where $u^{\prime }$ and $v^{\prime }$ are calculated according to:(3)$$\begin{array}{c} u^{\prime }=\frac{X^{\prime }}{Z^{\prime }}{f_{x}}^{\prime }+{u_{0}}^{\prime }\\ v^{\prime }=\frac{Y^{\prime }}{Z^{\prime }}{f_{y}}^{\prime }+{v_{0}}^{\prime }. \end{array}$$

The intrinsic parameters of the IR camera of the Kinect depth sensor and the colour camera and their extrinsic parameters are determined using a stereo camera calibration method. In the example shown in Figure 4a, there are many pixels in Regions 1 and 2 that have a value of zero because the aligned depth Regions 1 and 2 are larger than their corresponding Regions 1 and 2 in Figure 4a, and these pixels do not obtain depth values from Figure 4a. A dilation operation is used to recover the depth value of these pixels. A 3 × 3 rectangular structuring element is used to dilate the source depth map to determine the shape of a pixel’s neighbourhood over which the maximum is taken, according to:(4)$$dst\left(x,y\right)=\max_{\left(x^{\prime },y^{\prime }\right):element\left(x^{\prime },y^{\prime }\right)\neq 0}src\left(x+x^{\prime },y+y^{\prime }\right).$$

#### 3.1.2. Depth Map Hole Filling

From the aligned depth map (Figure 4c), there still exist a number of black holes that are labelled with green-coloured ellipses, and the largest hole labelled with “3” in green colour. These holes are caused by the structured light that the IR projector of the Kinect depth sensor emits, which was reflected in multiple directions, encountered transparent objects, and scattered from object surfaces [33]. To avoid incorrect segmentation, these depth holes must be filled.

The task is to use valid depth values around depth holes to fill the depth holes. Vijayanagar et al. [34] proposed a multi-resolution anisotropic diffusion (AD) method, which uses the colour image to diffuse the depth map and requires this process to be iterated many times in the multi-resolutions of the colour image for each resolution. Differently, as discussed in the next sub-section on depth map segmentation, the depth value of a filled hole only needs to be within the DoF at its neighbouring valid depth value. Therefore, the AD method is applied more efficiently in our work. (1) The AD filter is only applied to the depth map of the original size. (2) The conduction coefficients are only computed from the depth map. (3) Only one iteration of AD is applied because after one iteration, the differences between the depth value of the recovered pixel and its neighbours become less than the DoF at the recovered depth value, and thus, incorrect segmentation is avoided.

For an image *I*, the discrete form of the anisotropic diffusion equation, according to [35], is:(5)$$\begin{array}{cc} I_{\left(i,j\right)}^{t+1} & =I_{\left(i,j\right)}^{t}+\lambda (C_{N}\cdot d_{N}+C_{S}\cdot d_{S}\\ & +C_{W}\cdot d_{W}+C_{E}\cdot d_{E}{)}_{\left(i,j\right)}^{t}, \end{array}$$
where 0≤λ≤0.25 for the equation to be stable, *t* indicates the current iteration, *d* represents the depth value difference between the pixel $I_{\left(i,j\right)}$ and one of its four neighbours, and the subscripts *N*, *S*, *E*, and *W* denote the neighbouring pixels to the north, south, east, and west. The conduction coefficient *C* is:(6)$$C=g\left(d\right)=e^{\left(-{\left(d/K\right)}^{2}\right)},$$
where *K* is the standard deviation.

To recover the depth value of $I_{\left(i,j\right)}$, since the IR projector is located on the right side of Kinect and the IR camera is on the left side, the main depth holes (Region 3 in Figure 4c) are always to the left of an object, and we replace $I_{\left(i,j\right)}$ with $I_{\left(i-2,j\right)}$ to fill the depth holes. Thus, (Equation 5) is rewritten as:(7)$$\begin{array}{cc} I_{\left(i,j\right)} & =I_{\left(i-2,j\right)}+\lambda (C_{N}\cdot d_{N}+C_{S}\cdot d_{S}\\ & +C_{W}\cdot d_{W}+C_{E}\cdot d_{E}{)}_{\left(i-2,j\right)}, \end{array}$$
where:(8)$$\begin{array}{c} d_{N}=I_{\left(i-3,j\right)}-I_{\left(i-2,j\right)},\\ d_{S}=I_{\left(i-1,j\right)}-I_{\left(i-2,j\right)},\;\\ d_{W}=I_{\left(i-2,j-1\right)}-I_{\left(i-2,j\right)},\\ d_{E}=I_{\left(i-2,j+1\right)}-I_{\left(i-2,j\right)}.\; \end{array}$$

The aligned depth map after hole filling is shown in Figure 4d.

### 3.2. Graph-Based Depth Map Segmentation

After preprocessing the depth map, the depth map is segmented into distinct image block regions. Each segmented region must satisfy the DoF rule, as described below, to ensure all objects in this region appear in focus. In Figure 5, scene point *L* is at a distance of $u_{l}$ from the lens, *M* is at a distance of *u*, and *S* is at a distance of $u_{s}$. The three points are imaged as *l* at a distance of v${}_{l}$, *m* at a distance of v, and *s* at a distance of v${}_{s}$. Among the three scene points, only *M* is imaged in perfect focus at the image detector; *L* and *S* are imaged as a blurred circle with diameter δ centred around *m*. The DoF consists of two parts, the back DoF (b_DoF) and front DoF (f_DoF), and their values at a distance of *u* can be derived as:(9)$$b\_DoF\left(u\right)=u_{l}-u=F\delta u^{2}/\left(f^{2}-F\delta u\right)$$
(10)$$f\_DoF\left(u\right)=u-u_{s}=F\delta u^{2}/\left(f^{2}+F\delta u\right)$$
where F=f/d is the aperture value. Let Min and Max represent the minimum and maximum depth values in a segmented region, respectively, and let Diff represent the difference between Min and Max (i.e., Diff = Max − Min). Let b_DoF(Min) represent the back DoF when the camera is in focus at Min, f_DoF(Max) represent the front DoF when the camera is in focus at Max, and MaxDoF represent the larger value between b_DoF(Min) and f_DoF(Max). To ensure all objects in a segmented region all appear focused, the DoF rule requires that Diff be smaller than MaxDoF (i.e., Diff < MaxDoF).

In graph theory-based segmentation algorithms, a graph with vertices, image pixels, and edges corresponding to pairs of neighbouring vertices is established. Each edge has a weight initialized by the difference between the values of pixels on each side of the edge. In existing graph theory-based segmentation algorithms, blocks of pixels with low variability tend to be segmented into a single region. For an object with a wide depth range, the entire object crosses multiple DoFs and cannot appear focused in one focus setting. In this case, standard graph-based segmentation algorithms would incorrectly segment the entire object into a single region. For objects within a specific DoF of the camera, but with different depth values, the standard graph-based segmentation algorithms may unnecessarily segment these objects into different regions.

Figure 6a is the depth map of a real scene with its corresponding colour image shown in Figure 6b). We first applied the classic graph-based segmentation algorithm (Felz algorithm) [36], which segmented the depth map into three regions (Figure 6c). Table 1 summarizes the values of Min, Max, Diff, and MaxDoF of each segmented region. For Region 3, Diff is larger than MaxDoF, indicating that all the objects in Region 3 cannot appear focused in one focus setting. Since the depth values in Region 3 change gradually from 832 mm–1360 mm, they were incorrectly segmented into a single region. For Regions 1 and 2, when the camera was set to focus at the minimum depth value in Region 2 (2417 mm), and b_DoF was 1132 mm, which is larger than the difference (698 mm) between the minimum depth value in Region 2 (2417 mm) and the maximum depth value in Region 1 (3115 mm), indicating that the objects in Regions 1 and 2 can appear focused in one focus setting. In summary, with the Felz algorithm, Regions 1 and 2 in Figure 6c were unnecessarily segmented into two regions, and Region 3 in Figure 6c was incorrectly regarded as a single region.

In our depth map segmentation, a graph-based representation of the depth map is first established, in which pixels are nodes and edge weights measure the dissimilarity between nodes (e.g., depth differences). Given two components, $C_{1}$ and $C_{2}$, let min and max represent the minimum and maximum depth values among all the depth pixels within $C_{1}$ and $C_{2}$, diff equal max minus min, and b_DoF(min) and f_DoF(max) represent the back DoF and front DoF when the camera is set to focus at min and max, respectively. To ensure that the final segmented regions can all appear focused in one focus setting of the camera, we then impose the rule of DoF, i.e., only if diff is less than the larger value of b_DoF(min) and f_DoF(max) can the two components be merged.

The segmentation result using the proposed graph-based depth map segmentation algorithm is shown in Figure 6d. The Min, Max, Diff, and MaxDoF values of each segmented region are shown in Table 2. It can be seen that in every region, Diff is less than MaxDoF, indicating that all objects within each region can appear focused in one focus setting.

### 3.3. Construct All-in-Focus Image

Based on the segmented regions on the depth map, a focus algorithm is guided to locate an in-focus image for each region from among the multi-focus images captured at different focus settings. In our work, the focus algorithm of the normalized variance (NV) is used due to its best overall performance in terms of accuracy, number of false maxima, and noise level [37]. Consider a grey image *I* of size M×N, where *M* equals the number of rows and *N* is the number of columns. NV is computed according to:(11)$$NV=\frac{1}{M\times N\times \mu }\sum_{M}{\sum_{N}\left(I(x,y)-\mu \right)}^{2},$$
where μ is the mean grey value of image *I* and I(x,y) is the grey value of the pixel at position (x,y) of image *I*.

## 4. Experiments

### 4.1. Evaluation Metrics

Seven representative fusion methods were selected for comprehensive comparisons with our proposed method. These five methods are discrete wavelet transform (DWT) [38], nonsubsampled contourlet transform (NSCT) [39], image matting (IM) [18], guided filtering (GF) [17], spatial frequency-motivated pulse coupled neural networks in the nonsubsampled contourlet transform domain (NSCT-PCNN) [22], dense SIFT (DSIFT) [24], and the deep convolutional neural network (DCNN) [23]. DWT and NSCT are multi-scale transform methods; IM and GF are spatial methods; NSCT-PCNN is a PCNN-based and multi-scale transform method; DSIFT is a feature space method; and DCNN is a deep learning method. The source codes of these algorithms were obtained online (see the Appendix A).

In image fusion applications, there is a lack of a reference image or a fused image as the ground truth for comparing different algorithms. As reported in [25], fusion metrics are categorized into four groups: (1) information theory-based metrics, (2) image feature-based metrics, (3) image structural similarity-based metrics, and (4) human perception-inspired fusion metrics. In the experiments, six fusion metrics covering all four categories were chosen, including normalized mutual information $Q_{MI}$ [40], nonlinear correlation information entropy $Q_{NCIE}$ [41], gradient-based fusion metric $Q_{G}$ [42], phase congruency-based fusion metric $Q_{P}$ [25], Yang’s fusion metric $Q_{Y}$ [43], and the Chen–Blum metric $Q_{CB}$ [44]. $Q_{MI}$ and $Q_{NCIE}$ are information theory-based metrics; $Q_{G}$ and $Q_{P}$ are image feature-based metrics; $Q_{Y}$ is an image structural similarity-based metric; and $Q_{CB}$ is a human perception-based metric. These six fusion metrics were implemented using the image fusion evaluation toolbox at https://github.com/zhengliu6699. For all six metrics, a larger value indicates a better fusion result.

### 4.2. Source Images

Multi-focus source images from five different scenes were captured and used in this study (Appendix A). Figure 7 shows the source images of one of the scenes. Figure 7a is the depth map of the scene. The depth map segmentation resulted in only two regions: the front region and the background region, as shown in Figure 7b. Thus, the focus algorithm was guided to locate the two multi-focus source images (Figure 7c,d), which were then used to construct an all-in-focus image. The scenes tested in this work were intentionally set to have only two regions, and there were only two multi-focus source images because the online image fusion evaluation toolbox (https://github.com/zhengliu6699) was designed to evaluate the fusion performance of two source images. In addition, all the source codes of different multi-focus image fusion methods were also designed to fuse two images. The source images of other scenes are provided in the Appendix A and can be downloaded from the author’s GitHub website (https://github.com/robotVisionHang).

### 4.3. Comparison Results

The assessment metric values of the all-in-focus images constructed using our proposed method and other multi-focus image fusion algorithms for different scenes are summarized in Table 3. For each metric, the numbers in parentheses denote the score of each of the seven methods. The highest score was seven, and the lowest score was one. The higher the score, the better the method.

Table 4 shows the number of times of each method receiving a score, the total score of each method, and the overall ranking of the eight methods. Among the eight methods, our proposed method received a score of eight for the highest number of times and had the highest overall ranking. The results also reveal that our proposed method, DCNN, DSIFT, and IM outperformed GF, NSCT-PCNN, DWT, and NSCT, and GF performed better than other multi-scale transform methods (NSCT-PCNN, DWT, and NSCT).

The core process of state-of-the-art RGB-based multi-focus image fusion methods (e.g., DCNN, DSIFT, GF, and IM) is to compute a weight map by comparing the relative clearness level of multi-focus source images based on the deep convolutional neural network, dense SIFT feature, guided filter, and image matting, respectively. In our proposed method, the weight map was generated through segmenting the depth map. Take *A* and *B* as two multi-focus source images, and *W* is the weight map. A fused image, *F*, is constructed according to:(12)F=(1.0−W)∗A+W∗B,
where ∗ is an operation of pixel-wise multiplication. The range of values for *W* is 0.0–1.0. In a position (i,j) within *W*, a value of 0.0 means the fusion method judges that *A* is definitely clearer than *B*, and a value of 1.0 means *B* is definitely clearer than *A* in (i,j). If the fusion method is uncertain about whether *A* is definitely clearer than *B*, it assigns a value between 0.0 and 1.0 to represent the clearness level of *A* compared with *B*. A value less than 0.5 indicates that *A* is considered to be probably clearer than *B*; a value of 0.5 indicates that *A* and *B* are considered to be equally clear; and a value higher than 0.5 indicates that *B* is considered to be probably clearer than *A*.

The better performance of our proposed method compared to the other multi-focus image fusion methods can be understood by examining the weight maps they generated. For GF, the weight map for the detail layer was used to reconstruct the base layer and the detail layer of the fused image due to its more detailed reflection of the level of sharpness compared with the weight map for the base layer. Interestingly, the fused image reconstructed only with a detail layer (vs. with both base layer and detail layer [17]) generally obtained a higher score (see Appendix A).

The values in the weight map of DSIFT can take on 0.0, 0.5, or 1.0, and for DCNN, IM and GF, the values ranged from 0.0–1.0. In our proposed method, the weight map was generated through the segmented regions. For a scene with only two segmented regions, the values in the weight map within a segmented region were all zeros, since the pixels of one multi-focus source image within this region were considered in best focus. Similarly, the values in the weight map within the other segmented region were all ones.

To fuse the multi-focus source images shown in Figure 7c,d, the weight maps generated by our proposed method, DCNN, DSIFT, IM, and GF are shown in Figure 7. The weight maps of other test scenes can be found in the Appendix A. This scene only contains two regions, the front region and the background region. During image capturing, the distance from the front region and the background region was set to be sufficiently large to ensure that when one region is in focus, the other region is defocused. Figure 7e shows that the white front region and black background region are completely separated; the weight values in the front region are all ones, and the weight values in background region are all zeros, accurately reflecting the sharpness level of this scene. However, in Figure 7f–i, it can be seen that none of the DCNN, DSIFT, IM, and GF methods were able to generate a weight map as clean as the weight map generated by our proposed method because they rely on the colour information of the multi-focus source images for computing weight maps, which is susceptible to lighting, noise, and the texture of objects. Differently, our proposed method circumvents these limitations by making use of the depth map to directly determine weight maps.

The time consumption for constructing an all-in-focus image using our proposed method and other multi-focus image fusion algorithms was also quantified and compared. The sizes of the multi-focus source images and depth maps were 640×480. Tests were conducted on a computer with a 4-GHz CPU and 32 GB of RAM. The time consumption of our proposed method reported in Table 5 includes preprocessing the depth map, segmenting the depth map, and selecting in-focus images from multi-focus source images to construct an all-in-focus image. Our method took 33 ms on average to construct an all-in-focus image, among which preprocessing the depth holes cost 5 ms, segmenting the depth map cost 27.5 ms, and selecting in-focus images to construct the all-in-focus image cost 0.5 ms. The significantly lower time consumption of our method, compared to the RGB-based methods (see Table 5), is due to the low computational complexity stemming from the assistance of the depth map. Note that in practice, there are usually more than two multi-focus source images to be used to construct an all-in-focus image of a scene, and in accordance, the time consumption of other multi-focus image fusion methods increases linearly. Differently, for the proposed all-in-focus imaging method, since the time cost of preprocessing and segmenting the depth map is linear to the size of the depth map [36], as long as the size of the depth map from the depth sensor is fixed, the time cost of preprocessing and segmenting the depth map stays constant. Although the time cost of selecting in-focus images is linear to the number of multi-focus source images, due to its low computational complexity, the time consumption of our proposed method does not increase significantly when the number of multiple multi-focus source images becomes higher.

The proposed multi-focus image fusion method is highly dependent on the depth map from the depth sensor. Presently, the range of the Kinect depth sensor is limited to 0.5 m–5 m. However, the proposed method is not dependent on a specific depth sensor. For instance, the ZED stereo camera has a significantly larger operating range (0.5 m–20 m) and can obtain depth maps with a size up to 4416 × 1242 at 15 fps. Figure 8 shows the use of the ZED stereo camera for obtaining the depth map of more complex nature scenes.

## 5. Conclusions

This paper reported an efficient multi-focus image fusion method assisted by depth sensing. The depth map from a depth sensor was segmented with a modified graph-based segmentation algorithm. The segmented regions were used to guide a focus algorithm to locate an in-focus image for each region from among multi-focus images. The all-in-focus image was constructed by combining the in-focus images selected in each segmented region. The experimental results demonstrated the advantages of the proposed method by comparing the method with other algorithms in terms of six fusion metrics and time consumption. The proposed method enables the construction of an all-in-focus image within 33 ms and provides a practical approach for constructing high-quality all-in-focus images that can potentially be used as reference images.

## Figures and Tables

**Figure 1 sensors-19-01409-f001:**
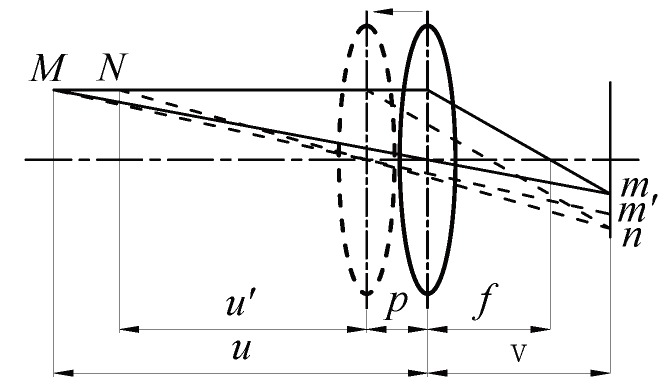
A scene point *M* at a distance of *u* from the lens is imaged in focus by an image detector at a distance of vfrom the lens with a focal length *f*. If the lens moves forward with a distance of *p*, *M* is imaged as a blurred circle around $m^{\prime }$, while the near scene point *N* at a distance of $u^{\prime }$ from the lens is imaged in focus at *n*.

**Figure 2 sensors-19-01409-f002:**
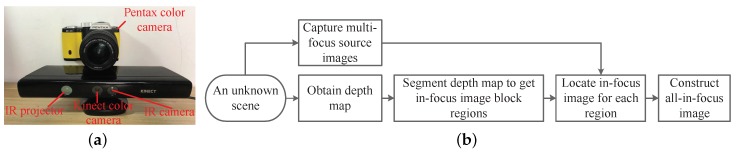
(**a**) Setup used in this work for evaluating the proposed multi-focus image fusion method. (**b**) Flowchart of the method.

**Figure 3 sensors-19-01409-f003:**
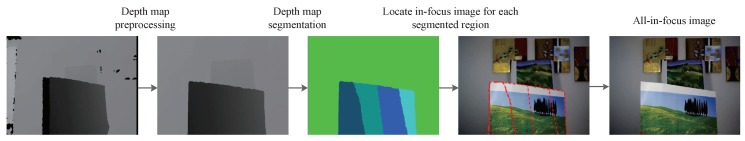
Main steps and intermediate results when applying the proposed multi-focus image fusion method to construct an all-in-focus image of a real scene.

**Figure 4 sensors-19-01409-f004:**
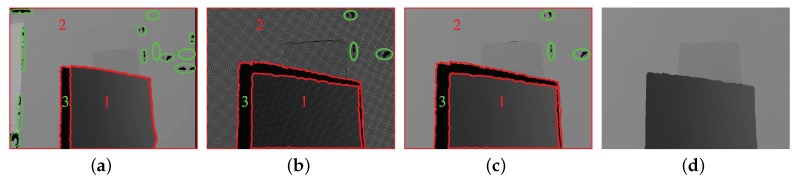
Depth map preprocessing: (**a**) raw depth map from the Kinect depth sensor, (**b**) aligned depth map, (**c**) aligned depth map after dilation, and (**d**) aligned depth map after hole filling. Black holes are labelled with green-coloured ellipses; the largest hole is labelled with “3” in green colour; the front object region and background region are labelled with “1” and “2”, respectively.

**Figure 5 sensors-19-01409-f005:**
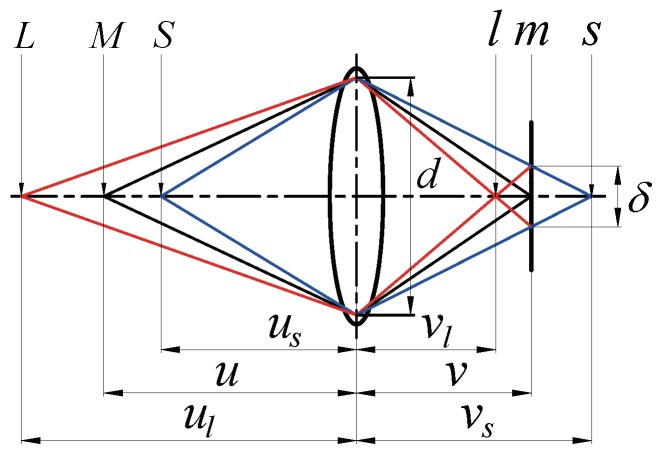
Diagram for calculating the depth of field (DoF).

**Figure 6 sensors-19-01409-f006:**
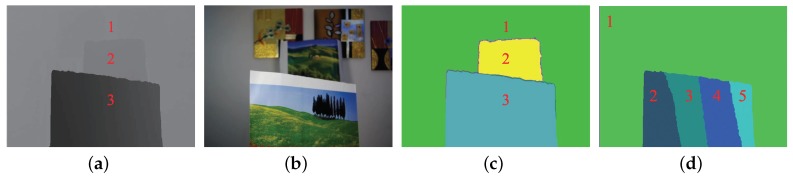
Depth map segmentation using the standard graph-based segmentation algorithm and our modified graph-based segmentation algorithm. (**a**) Raw depth map of a real scene. (**b**) Colour image of the scene. (**c**) Segmentation result by using the standard graph-based Felz algorithm. (**d**) Segmentation result using our modified algorithm.

**Figure 7 sensors-19-01409-f007:**
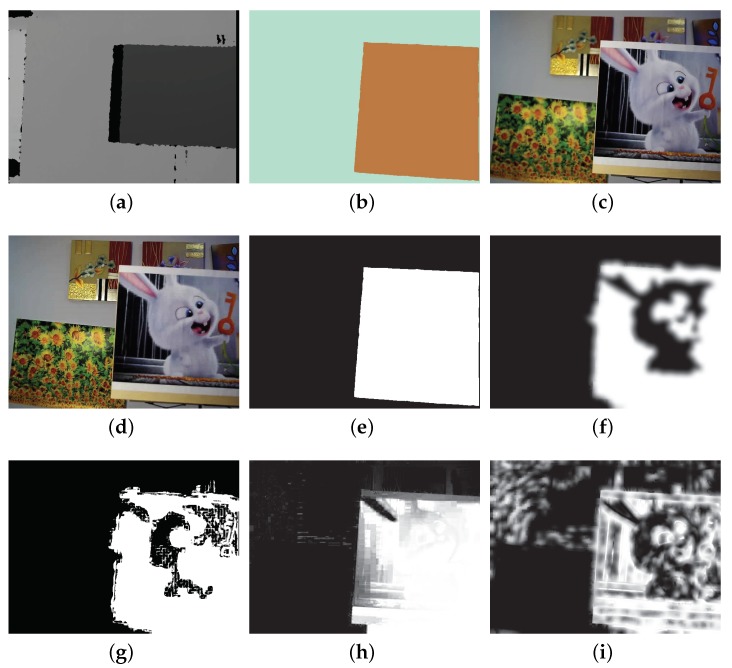
Images of one test scene (**a**–**d**) and weight maps of different multi-focus image fusion methods (**e**–**i**). This is the fifth scene in Appendix A. (**a**) Depth map of the scene. (**b**) Segmentation result of the depth map. (**c**) Multi-focus source image with the front object in best focus. (**d**) Multi-focus source image with the background objects in best focus. Weight maps generated by (**e**) our proposed method, (**f**) DCNN, (**g**) dense SIFT (DSIFT), (**h**) image matting (IM), and (**i**) guided filtering (GF). These weight maps are also shown in the fifth group of weight maps in Appendix A.

**Figure 8 sensors-19-01409-f008:**
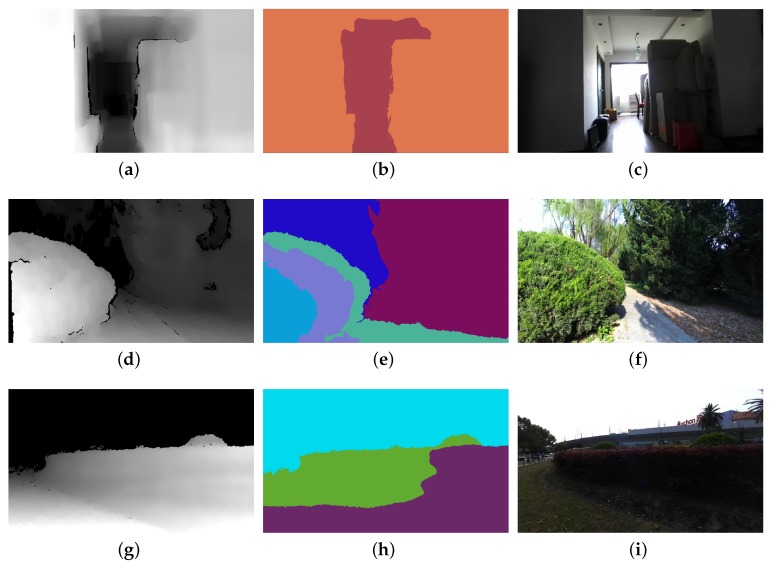
(**a**,**d**,**g**) Depth maps obtained via the use of a ZEDstereo camera. (**b**,**e**,**h**) In-focus image block regions determined by segmenting depth maps. (**c**,**f**,**i**) Corresponding all-in-focus colour images.

**Table 1 sensors-19-01409-t001:** Depth values (in mm) of segmented regions in Figure 6c.

Region	Min	Max	Diff	MaxDoF	Diff<MaxDoF?
1	2722	3115	393	1526	Yes
2	2417	2639	222	1132	Yes
3	832	1360	528	207	No

**Table 2 sensors-19-01409-t002:** Depth values (in mm) of segmented regions in Figure 6d.

Region	Min	Max	Diff	MaxDoF	Diff<MaxDoF?
1	2463	3140	677	1186	Yes
2	855	962	107	109	Yes
3	950	1085	135	136	Yes
4	1088	1269	181	182	Yes
5	1273	1412	139	257	Yes

**Table 3 sensors-19-01409-t003:** Quantitative assessments of the proposed all-in-focus imaging method and other existing multi-focus image fusion methods. Parentheses denote the scores of a method when compared with the other six methods. The higher the score, the better the method. Eight is the highest score, and one is the lowest score. NSCT-PCNN, pulse coupled neural networks in the nonsubsampled contourlet transform domain.

Scenes	Metrics	Methods							
		DWT	NSCT	IM	GF	NSCT-PCNN	DSIFT	DCNN	Ours
1	$Q_{MI}$	1.1478(2)	1.0451(1)	1.3869(5)	1.3402(4)	1.3372(3)	1.4235(8)	1.3903(6)	1.4201(7)
	$Q_{NCIE}$	0.8463(2)	0.8408(1)	0.8629(4)	0.8597(3)	0.8646(6)	0.8681(8)	0.8635(5)	0.8653(7)
	$Q_{G}$	0.6694(3)	0.4408(1)	0.6998(5)	0.6946(4)	0.6421(2)	0.7079(6)	0.7094(7)	0.7153(8)
	$Q_{P}$	0.8344(2)	0.7255(1)	0.9129(8)	0.9023(4)	0.8516(3)	0.9049(5)	0.9112(7)	0.9099(6)
	$Q_{Y}$	0.8992(2)	0.7262(1)	0.9548(5)	0.9412(4)	0.9275(3)	0.9710(6)	0.9721(7)	0.9766(8)
	$Q_{CB}$	0.7372(2)	0.6935(1)	0.7688(5)	0.7634(4)	0.7977(8)	0.7575(3)	0.7708(6)	0.7742(7)
2	$Q_{MI}$	0.9504(2)	0.8125(1)	1.2323(7)	1.1674(4)	1.0457(3)	1.2308(6)	1.2250(5)	1.2504(8)
	$Q_{NCIE}$	0.8308(2)	0.8250(1)	0.8480(7)	0.8426(4)	0.8374(3)	0.8468(6)	0.8465(5)	0.8489(8)
	$Q_{G}$	0.6387(3)	0.3889(1)	0.6855(6)	0.6747(4)	0.5777(2)	0.6834(5)	0.6879(7)	0.6954(8)
	$Q_{P}$	0.8273(3)	0.6922(1)	0.9159(5)	0.9175(6)	0.8269(2)	0.9141(4)	0.9206(8)	0.9191(7)
	$Q_{Y}$	0.9012(3)	0.6908(1)	0.9655(6)	0.9431(4)	0.8976(2)	0.9627(5)	0.9716(7)	0.9832(8)
	$Q_{CB}$	0.7231(2)	0.6681(1)	0.7856(6)	0.7627(3)	0.7744(4)	0.7832(5)	0.7887(7)	0.7977(8)
3	$Q_{MI}$	0.9101(2)	0.8422(1)	1.1820(5)	1.1500(4)	1.0052(3)	1.2015(7)	1.1927(6)	1.2089(8)
	$Q_{NCIE}$	0.8284(2)	0.8255(1)	0.8437(5)	0.8414(4)	0.8344(3)	0.8448(7)	0.8442(6)	0.8454(8)
	$Q_{G}$	0.6608(3)	0.4649(1)	0.7039(5)	0.6998(4)	0.5672(2)	0.7079(6)	0.7099(7)	0.7143(8)
	$Q_{P}$	0.8266(3)	0.7660(1)	0.9070(5)	0.9115(7)	0.8053(2)	0.9112(6)	0.9127(8)	0.9033(4)
	$Q_{Y}$	0.9151(3)	0.7796(1)	0.9742(5)	0.9602(4)	0.8834(2)	0.9759(6)	0.97997	0.9825(8)
	$Q_{CB}$	0.7059(2)	0.6699(1)	0.7816(5)	0.7681(4)	0.7169(3)	0.7903(6)	0.7949(7)	0.7954(8)
4	$Q_{MI}$	0.8384(2)	0.7653(1)	1.1384(5)	1.0978(4)	0.9426(3)	1.1727(7)	1.1520(6)	1.1828(8)
	$Q_{NCIE}$	0.8249(2)	0.8220(1)	0.8408(5)	0.8382(4)	0.8310(3)	0.8430(7)	0.8415(6)	0.8439(8)
	$Q_{G}$	0.6269(3)	0.4355(1)	0.6738(5)	0.6642(4)	0.5434(2)	0.6786(6)	0.6822(7)	0.6886(8)
	$Q_{P}$	0.7967(3)	0.7586(2)	0.8972(4)	0.9039(6)	0.7443(1)	0.9020(5)	0.9048(7)	0.9067(8)
	$Q_{Y}$	0.9047(3)	0.7491(1)	0.9692(5)	0.9500(4)	0.8729(2)	0.9777(6)	0.9837(7)	0.9890(8)
	$Q_{CB}$	0.6908(2)	0.6486(1)	0.7713(5)	0.7527(4)	0.7075(3)	0.7828(6)	0.7852(8)	0.7834(7)
5 (Figure 7)	$Q_{MI}$	0.9352(2)	0.8659(1)	1.1746(5)	1.1420(4)	0.9868(3)	1.2248(7)	1.1968(6)	1.2311(8)
	$Q_{NCIE}$	0.8305(2)	0.8276(1)	0.8444(5)	0.8435(4)	0.8335(3)	0.8481(7)	0.8465(6)	0.8482(8)
	$Q_{G}$	0.6432(3)	0.4472(1)	0.6720(5)	0.6594(4)	0.5506(2)	0.6751(6)	0.6753(7)	0.6885(8)
	$Q_{P}$	0.8381(3)	0.7649(1)	0.9011(7)	0.8953(4)	0.7858(2)	0.8973(5)	0.8984(6)	0.9214(8)
	$Q_{Y}$	0.9016(3)	0.7483(1)	0.9628(5)	0.9419(4)	0.8702(2)	0.9698(6)	0.9769(7)	0.9802(8)
	$Q_{CB}$	0.7117(2)	0.6785(1)	0.7860(5)	0.7607(4)	0.7186(3)	0.7966(7)	0.7964(6)	0.8014(8)

**Table 4 sensors-19-01409-t004:** Scores and rankings of the methods.

		Scores	8	7	6	5	4	3	2	1	Total Scores	Ranking
	Number of Times											
Methods												
Ours			23	5	1	0	1	0	0	0	229	1
DCNN			3	14	10	3	0	0	0	0	197	2
DSIFT			2	7	13	6	1	1	0	0	180	3
IM			1	3	3	21	2	0	0	0	160	4
GF			0	1	2	0	25	2	0	0	125	5
NSCT-PCNN			1	0	1	0	1	14	12	1	85	6
DWT			0	0	0	0	0	0	13	17	73	7
NSCT			0	0	0	0	0	0	1	29	31	8

**Table 5 sensors-19-01409-t005:** Running time (seconds) of the proposed method and existing multi-focus image fusion algorithms for the five test scenes.

Scenes	Methods							
	DWT	NSCT	IM	GF	NSCT-PCNN	DSIFT	DCNN	Ours
1	0.2054	35.7285	3.2084	0.3351	243.2443	8.8385	132.9873	0.030
2	0.2031	35.5960	3.1097	0.3491	243.8029	11.4488	131.7024	0.035
3	0.2061	35.7128	2.9816	0.3473	243.4221	7.6047	131.6626	0.033
4	0.2039	35.7426	2.9719	0.3457	243.8831	7.3378	127.3014	0.032
5 (Figure 7)	0.2050	35.7939	2.9131	0.3452	243.1754	9.4629	132.2269	0.035
Average	0.2047	35.7148	3.0369	0.3445	243.5056	8.9385	131.1761	0.033

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
