# Peer review of "Construction of All-in-Focus Images Assisted by Depth Sensing"

_sensors, 2019, doi:10.3390/s19061409_

Round 1
Reviewer 1 Report
This work presents a method to combine a depth sensing device with multi-focus fusion with a camera, to provide good estimates of image depth and produce images that are in focus at all depths. The authors compare to multi-focus fusion approaches without the added
depth sensor, and conclude that their approach (which has more
information) works better. Although I am not aware of this exact approach being used before, there is a wealth of work combining information from depth sensing devices and cameras for 3D imaging. None of this related work is mentioned or cited. That previous work has already demonstrated the advantages of using a depth sensor for depth estimation, as reported here. It is not clear to me if the current approach extracts the same information as those 3D imaging approaches, or if the multi-focus fusion provides some advantage in resolution. I think this manuscript needs to discuss this, and should compare the proposed approach to these existing methods that combine information from cameras and depth sensors.
Some randomly selected examples
(Single camera sensing with depth sensor)
Yang Q, Yang R, Davis J, Nistér D. Spatial-depth super resolution for range images. In Computer Vision and Pattern Recognition, 2007. CVPR'07. IEEE Conference on 2007 Jun 17 (pp. 1-8). IEEE.
S Zhang, C Wang, S Chan, A new high resolution depth map estimation system using stereo vision and kinect depth sensing. Journal of Signal Processing Systems. 2015 Apr 1;79(1):19-31.
Author Response
Dear Editor and Reviewers,
We thank you for reviewing our manuscript. We value your insightful comments and have revised the manuscript accordingly. Every comment has been carefully addressed, as the attached PDF file.

Reviewer 2 Report
This paper presents a method to generate an *all-in-focus* image assited by depth information. The paper is well written and results seem convincing. However, I have two comments/concerns for the authors:
1) Are the authors saying that their method is the first one to propose an *all-in-focus* image generation strategy using information from the depth images? I have not seen any state-of-the-art analysis made by the authors in this sense in the Introduction (or anywhere else in the paper).
2) I found it very hard to understand how an *all-in-focus* image is generated, starting from the segmentation of the depth images. I do not understand how *Min*, *Max*, *Diff*, *MaxDoF*, etc., in pages 6 and 7 (and Table 1 and 2) are used and combined together to obtain an *all-in-focus* image. Could the authors please let me know how they are able to dentify and make sure they have an *in-focus* region starting from the results of the segmented depth map images?
Author Response

(The authors gave the same response as above.)

Round 2
Reviewer 2 Report
I am satisfied with the changes proposed by the authors and therefore also happy to accept it in its present form.